# High Performance and Self-Humidifying of Novel Cross-Linked and Nanocomposite Proton Exchange Membranes Based on Sulfonated Polysulfone

**DOI:** 10.3390/nano12050841

**Published:** 2022-03-02

**Authors:** Xinyu Li, Zhongxin Zhang, Zheng Xie, Xinrui Guo, Tianjian Yang, Zhongli Li, Mei Tu, Huaxin Rao

**Affiliations:** College of Chemistry and Materials Science, Jinan University, Guangzhou 510632, China; a1563927622@163.com (X.L.); jnuzwb2022@163.com (Z.Z.); xiez8700778@163.com (Z.X.); guoxinrui99@163.com (X.G.); jnusirui@163.com (T.Y.); oytxm@stu2020.jnu.edu.cn (Z.L.); tumei@jnu.edu.cn (M.T.)

**Keywords:** fuel cells, proton exchange membrane, polymer brush, sulfonated polysulfone, graphene oxide, self-humidifying, nanoparticles

## Abstract

The introduction of inorganic additive or nanoparticles into fluorine-free proton exchange membranes (PEMs) can improve proton conductivity and have considerable effects on the performance of polymer electrolyte membrane fuel cells. Based on the sol–gel method and in situ polycondensation, novel cross-linked PEM and nanocomposite PEMs based on a sulfonated polysulfone (SPSU) matrix were prepared by introducing graphene oxide (GO) polymeric brushes and incorporating Pt-TiO_2_ nanoparticles into an SPSU matrix, respectively. The results showed that the incorporation of Pt-TiO_2_ nanoparticles could obviously enhance self-humidifying and thermal stability. In addition, GO polymer brushes fixed on polymeric PEM by forming a cross-linked network structure could not only solve the leakage of inorganic additives during use and compatibility problem with organic polymers, but also significantly improve proton conductivity and reduce methanol permeability of the nanocomposite PEM. Proton conductivity, water uptake and methanol permeability of the nanocomposite PEM can be up to 6.93 mS cm^−1^, 46.58% and be as low as 1.4157 × 10^−6^ cm^2^ s−1, respectively, which represent increases of about 70%, about 22% and a decrease of about 40%, respectively, compared with that of primary SPSU. Therefore, the synergic action of the covalent cross-linking, GO polymer brush and nanoparticles can significantly and simultaneously improve the overall performance of the composite PEM.

## 1. Introduction

Proton exchange membrane (PEM) are considered as the “heart” of the proton exchange membrane fuel cell (PEMFC), and its properties impact heavily on the performance, cost and even lifespan of PEMFC [1,2,3,4,5]. Perfluorosulfonic acid polymers such as Nafion are the most widely accepted and commercialized membranes to date that are used as PEM. However, conductivity loss at high temperatures, deficient membrane durability and methanol crossover through the membrane, which causes DMFC performance reduction, inhibit their use in large scale DMFC applications [6,7]. At present, the fluorine-free sulfonated aromatic PEMs have received tremendous attention due to their lower cost, higher hydrophilic ability and simpler preparation processing compared to that of the commercial Nafion [8,9,10,11]. Therefore, sulfonated polysulfone (SPSU) PEM, which are known as one of the most promising PEM without containing fluorine, are being widely studied because of excellent mechanical properties, good film-forming ability and low cost [12,13,14,15]. However, the thermal stability, chemical stability, water retention capacity and proton conductivity of the sulfonated polysulfone PEM should be further enhanced because of the low dissociation ability between carbon bonds and hydrogen bonds [16,17].

Sulfonated polysulfone PEMs only made of a single material can hardly meet the high requirements of PEMFC in all aspects [18,19]. So, the application of composite materials composed of two or more materials with multiple properties has been widely studied in recent years [20,21,22,23,24]. For example, the introduction of inorganic additives into polymeric PEMs is a simple and effective way to develop some properties of PEMFC under high temperature, low humidity or no humidity conditions. Some inorganic additives (SiO_2_, ZrO_2_, TiO_2_, Al_2_O_3_) and some natural ores (montmorillonite, zeolite) with great moisturizing property, which are added into polymeric membrane, can enhance obviously moisturizing property and further keep proton conductivity under high temperature or low humidity condition [25,26,27,28,29,30]. However, the introduction of these inorganic additives with special construction can only develop a moisturizing property and physicochemical stability, but not obviously improve proton conductivity. In addition, the leakage problem of inorganic small molecules for PEMs will inevitably occur in the process of long-term use, due to simple physical doping.

Hence, the organic–inorganic composite membrane materials have received considerable attention in order to fix small molecules into a polymer matrix [31,32,33,34]. One efficient method is that in which the network structures formed by inorganic small molecules are dispersed in the polymeric matrix [35,36]. For example, a variety of alkoxy silane or alkyl titanate will form a three-dimensional network structure inside the membrane through a sol–gel reaction, which can effectively improve the stability and moisture retention ability of the membrane. However, the dispersion method cannot completely solve the leakage of inorganic materials, and will bring the compatibility problem between inorganic materials and polymer matrix. Sun synthesized TiO_2_-SO_4_^2−^ solid superacid nanoparticles by the sol–gel method and compounded with Nafion. The proton conductivity of the composite membrane is slightly lower than that of pure recast Nafion membrane, but it has low methanol permeability [37]. Another efficient method is that in which inorganic additives are fixed in the polymer matrix by some interaction (such as chemical bond) [38,39]. Wu prepared hybrid acid–base polymer membranes by blending sulfonated poly(2,6-dimethyl-1,4-phenylene oxide) with (3-aminopropyl)triethoxysilane through a sol–gel process. The acid–base interaction improved not only the membrane homogeneity and thermal stability but also the mechanical strength and flexibility [40]. Of course, some special inorganic additives (such as phosphate) which form hardly network structure or chemical bonding with a polymeric matrix can form an acid–base interaction with an alkaline polymer matrix through static electricity [41,42]. Yu prepared high PA-doped paratactic PBI polymer membranes from PPA solution by sol–gel method. The composite membrane obtained by this method has a higher proton conductivity [43].

The aim of this study is developing a novel nanocomposite membrane using functionalized graphene oxide (GO) and Pt-TiO_2_ nanoparticles in order to solve the leakage of the inorganic additives and overcome the trade-off effect between proton conductivity and moisturizing property. Herein, a functionalized GO polymeric brushes (FPGO) was first synthesized by the sol–gel method and the hydrosilylation reaction and then was used to modify the SPSU matrix by in situ polycondensation. Then, Pt nanoparticles were deposited on TiO_2_ by redox method to form Pt-TiO_2_ nanoparticles. Finally, a novel nanocomposite PEM with the cross-linked network structures was fabricated by introducing GO polymer brush and incorporating Pt-TiO_2_ nanoparticles as additives into the modified SPSU matrix.

## 2. Materials and Methods

### 2.1. Materials

Titanium dioxide (50 nm), sodium borohydride, sulfonated polysulfone (SPSU, MW = 80,000, degree of sulfonation = 50%, distribution index = 2) were all obtained from Sigma-Aldrich. Graphene oxide (GO) was purchased from Suzhou Hengqiu Technology. Vinylmethyldimethoxysilane (VMDMO), 4-chlorostyrene, methanol and absolute ethanol were all provided from Adamas; 2,4,6,8-Tetramethylcyclotetrasiloxane (D_4_^H^) and chloroplatinic acid catalyst were supplied by Chengdu Organic Silicon Research, China. N,N-dimethylacetamide (DMAc), hydrogen peroxide, sodium chloride, and ferrous sulfate were all purchased from Guangzhou Chemical Reagent Factory. Nafion 117 was received from Shanghai Sanlu Industrial Co., Ltd.

### 2.2. Preparation of GO Polymeric Brushes

GO polymeric brushes, which were used as an additive for the nanocomposite membrane, were prepared by sol–gel method according to our previous research work [44]. In short, polymeric brushes-modified graphene oxide (PGO) was first synthesized by surface precipitation polymerization between vinylmethyldimethoxysilane (VMDMO) and GO nanosheets, at 65 °C, under magnetic stirring for 24 h. Then, D_4_^H^/4-chlorostyrene cyclic cross-linker was prepared by the hydrosilylation process between D_4_^H^ and 4-chlorostyrene at room temperature. Finally, functionalized GO polymeric brushes nanosheets (FPGO) were synthesized via hydrosilylation reaction between the cyclic cross-linker and PGO under rapid stirring and the catalytic action of chloroplatinic acid.

### 2.3. Preparation of Pt-TiO_2_ Nanoparticles

50 mg TiO_2_ was dispersed in 50 mL deionized water under ultrasonication for 2 h. An amount of 2 mL chloroplatinic acid solution was added into TiO_2_ dispersion liquid to form a mixture solution, followed by stirring vigorously at room temperature for 1 h. An amount of 0.2 M NaBH_4_ solution was added drop by drop to the mixed solution under ice-water bath, and then stirred at room temperature for 12 h. Afterward, the product was separated by centrifugation and then washed repeatedly with anhydrous ethanol for three times. The final product was dried in vacuum at 70 °C for 6 h to obtain the gray-black Pt-TiO_2_ nanoparticles.

### 2.4. Preparation of the Cross-Linked PEM and Nanocomposite PEMs

An amount of 2.0 g SPSU was dissolved in 8.5 mL DMAc under stirring at 40 °C. An amount of 20.0 mg FPGO was dispersed in 10.5 mL DMAc under ultrasonication for 2 h, then the FPGO suspension was added into the SPSU solution. Then, the mixed solution was stirred for 0.5 h to form the evenly solution and refluxed at 100 °C for 20 h. Through the condensation reaction between the -OK terminal groups of SPSU and the 4-chlorostyrene groups of FPGO, inorganic additives and polymer matrix can be cross-linked and FPGO/SPSU cross-linked membrane was prepared by solution-casting method. 

After the above FPGO/SPSU solution was cooled to room temperature, the prepared 2.0 mg Pt-TiO_2_ nanoparticles were added to the solution. After 0.5 h of ultrasonic treatment, the mixed solution was cast on the mold to form Pt-TiO_2_/FPGO/SPSU nanocomposite PEM. The membrane was dried at 40 °C for 12 h to remove the residual solvent, followed by successively drying at 90 °C for 12 h. After vacuum drying at 60 °C, the nanocomposite PEM with 0.1 wt.% Pt-TiO_2_ content was obtained. The prepared nanocomposite PEMs were designated as Pt-TiO_2_/FPGO/SPSU-X, where X was the weight ratio percentage of Pt-TiO_2_ nanoparticles to FPGO/SPSU. As a comparison, pristine SPSU membrane and FPGO/SPSU membranes were also fabricated.

### 2.5. Characterization

#### 2.5.1. Characterization of Pt-TiO_2_ Nanoparticles

The X-ray diffraction patterns were collected by X-ray diffractometer (XRD, MiniFlex, Japanese Science, Tokyo, Japan), with the data collection range of 5°~60° and the scanning speed of 5°/min. The size and distribution of Pt-TiO_2_ nanoparticles were determined by nano particle size and zeta potential analyzer (NANO ZS, Malvern, Malvern, UK). The morphology of Pt-TiO_2_ nanoparticles were observed by field emission transmission electron microscopy (TEM, JEOL 2100F, Japanese electron, Tokyo, Japan).

#### 2.5.2. The Morphology of the Nanocomposite PEM

Scanning electron microscopy (SEM, ULTRA 55, Germany ZEISS Company, Oberkochen, Germany) was used to observe the cross-sectional morphology of the membrane samples. Firstly, the membrane samples were frozen in liquid nitrogen, and then taken out and quickly broken. The membrane sample section was sprayed with gold under vacuum conditions.

#### 2.5.3. Measurement of Water Uptake (WU) and Swelling Ratio (SR)

After vacuum drying at 60 °C for 24 h, the weight of the membrane samples was measured to be *W_dry_* (g). Then, the membrane samples were immersed in deionized water for 48 h, and then removed. The water on the membrane surface was wiped off with filter paper, and the mass of the membrane samples were weighed as *W_wet_* (g). The water uptake (WU) of the membrane samples is calculated by Equation (1).

The membrane sample was cut into a square of 1 cm × 1 cm and the surface area was *A_dry_* (cm^2^). Then, the membrane samples were immersed in 20 mL deionized water and divided into four groups, which were stored at 30 °C, 50 °C, 70 °C and 90 °C for 24 h, respectively. After removal, the surface area of the membrane sample was *A_wet_* (cm^2^). The swelling ration (SR) of the membrane samples is calculated by Equation (2).
(1)$$\mathrm{Water}\;\mathrm{Uptake}\left(\%\right)=\frac{W_{wet}-W_{dry}}{W_{dry}}\times 100\%$$
(2)$$\mathrm{Swelling}\;\mathrm{Ration}\left(\%\right)=\frac{A_{wet}-A_{dry}}{A_{dry}}\times 100\%$$
where *W_dry_* and *A_dry_* are the weight and the surface area of the dry membranes, and *W_wet_* and *A_wet_* are the weight and the surface area of the wet membranes, respectively.

#### 2.5.4. Measurement of Ion Exchange Capacity (IEC)

Ion exchange capacity (IEC) of the membrane samples was determined by acid–base titration. Firstly, membrane samples were soaked in 1 M HCl solution and heated to 70 °C for 1 h, which changed them into the H^+^ form. The samples were then washed with distilled water several times to remove excess HCl, and right after that they were soaked in boiling water for 1 h, which confirmed the stability of membranes in hydrolytic conditions. In the next step, we took the samples in 50 mL of 1 M NaCl solution heated to 40 °C and equilibrated for at least 24 h to replace the protons by sodium ions. The remaining solution was titrated with 0.01 M *NaOH* solution using phenolphthalein as an indicator. IEC of the membrane samples is calculated by Equation (3).
(3)$$\mathrm{IEC}=\frac{C_{NaOH}\times V_{NaOH}}{W_{dry}}\times 100\%$$
where *W_dry_* is the weight of dry membrane (g), *V_NaOH_* is the volume of *NaOH* solution consumed by titration (mL), and *C_NaOH_* is the concentration of *NaOH* solution (mol/L).

#### 2.5.5. Measurement of Thermal Stability

The thermal properties of the nanocomposite membranes were evaluated by thermogravimetric analyzer (TGA, TG209F3, Netzsch, Serb, Germany) at 10 °C/min from 40 °C to 800 °C.

#### 2.5.6. Measurement of Mechanical Properties

The mechanical properties of the nanocomposite membrane were tested using a material tensile tester (SHMADZU AG-1, Shimadzu, Kyoto City, Japan). The tensile rate was 2 mm/min, and each sample was tested three times in parallel.

#### 2.5.7. Measurement of Oxidation Stability

The nanocomposite membrane was immersed in 20 mL Fenton reagent (2 mg/L FeSO_4_ in 3% H_2_O_2_), followed by placing in an oven at 80 °C to observe the decomposition of the samples with different times.

#### 2.5.8. Measurement of Proton Conductivity

Proton conductivity of the nanocomposite membrane was measured using an electrochemical workstation (Versa Stat 3, AMETEK, Shanghai, China). The test method was double-electrode AC impedance spectroscopy. The perturbation voltage was 10 mV and the test frequency was 1 Hz~10^5^ Hz. Proton conductivity of the membrane samples is calculated by Equation (4).
(4)$$\sigma =\frac{L}{R\times A}$$
where σ (S/cm) is the proton conductivity of the membranes, *L* (cm) is the distance between the two electrodes, *R* (Ω) is the impedance of the membrane samples measured under the AC impedance method, and *A* (cm^2^) is the cross-sectional area of the membrane samples.

#### 2.5.9. Measurement of Methanol Permeability and Selectivity

In the glass diffusion cell, the methanol permeability of various membrane samples was carried out at 30 °C, and the glass diffusion cell was composed of two compartments separated by proton exchange membrane. Prehydrate the membrane for 24 h and then clamp it between two diffusion chambers. The compartment *A* was first filled with 50 wt.% methanol aqueous solution, and the compartment *B* was first filled with deionized water. The methanol concentration in compartment *B* was measured immediately during the diffusion process using a digital refractometer (ABBE WYA 2WAJ, Shanghai, China). The methanol permeability of the composite membrane is calculated by Equation (5).
(5)$$P=\frac{SV_{B}L}{AC_{A}}$$
where *V_B_* is the volume of the solution in the *B* compartment (cm^3^), *L* is the thickness of the membrane sample (cm), *A* is the contact area between the membrane sample and the solution (cm^2^), and *C_A_* is the molar concentration of the methanol solution in the A compartment (mol/L), *S* is the slope of the molar concentration—time curve of methanol solution in compartment *B*.

The comprehensive performance of membranes was evaluated by selectivity (*β*, S s cm^−3^), which is defined as:(6)$$\beta =\frac{\sigma }{P}$$

## 3. Results

### 3.1. Preparation Process of the Cross-Linked PEM and the Nanocomposite PEMs

Graphene oxide (GO) representing 2D carbonaceous materials have received considerable attention, due to high performance [45]. It has been demonstrated that the protons can be transferred by the carboxylic acid groups along edges of GO and epoxy groups on the surface of GO [46]. Therefore, we develop a novel nanocomposite PEM by using functionalized GO and Pt-TiO_2_ nanoparticles in order to solve the leakage of the inorganic additives and overcome the trade-off effect between proton conductivity and moisturizing property.

The preparation process of FPGO and FPGO/SPSU is shown in Figure 1. Firstly, the cyclic silane cross-linked agent (D_4_^H^/4-chlorostyrene) containing active Si−H bond was synthesized by hydrosilylation of D_4_^H^ and p-chlorostyrene (4-chlorostyrene) with the molar ratio of 3:1 under the chloroplatinic acid catalysis. Then, linear VMDMO was grafted onto the surface of GO sheet by the sol–gel method to prepare the GO polymer brush with a sandwich structure. Secondly, the functional GO polymer brushes (FPGO) were prepared by addition reaction between D_4_^H^/4-chlorostyrene cross-linked agent and PGO polymer brushes containing double bond. Finally, FPGO/SPSU cross-linked PEM was prepared by in situ polycondensation of SPSU terminal OK group and chlorobenzene group (Cl) on FPGO surface.

In order to improve the self-humidifying performance of SPSU matrix membrane, Pt nanoparticles were first deposited on nano-TiO_2_ by redox method using NaBH_4_ as a catalyst to prepare Pt-TiO_2_ nanoparticles, followed by dispersing in FPGO/SPSU solution. Finally, Pt-TiO_2_/FPGO/SPSU nanocomposite PEMs with different Pt-TiO_2_ contents were prepared by impregnation method and solution casting method (shown in Figure 2).

### 3.2. Characterization of Pt-TiO_2_ Nanoparticles

The XRD patterns of Pt-TiO_2_ nanoparticles and nano TiO_2_ are shown in Figure 3a. In the XRD pattern of Pt-TiO_2_, the diffraction peak intensity of TiO_2_ is obviously weakened due to the incorporation of Pt, and the diffraction peaks of Pt (111), (200) and (220) crystal planes appear at 2*θ* = 40°, 2*θ* = 46° and 2*θ* = 67.5°, respectively. This indicates that Pt nanoparticles are successfully incorporated into nano TiO_2_ to form Pt-TiO_2_ nanoparticles. The average particle sizes of Pt-TiO_2_ nanoparticles (about 80 nm) measured by the nanoparticle size analyzer are larger than these of nano TiO_2_ (about 2 nm). In addition, the TEM images of Pt-TiO_2_ nanoparticles (Figure 3b) indicates further that Pt nanoparticles are uniformly dispersed on the surface of TiO_2_ particles and the interface between particles.

### 3.3. Characterization of the Cross-Linked PEM and the Nanocomposite PEMs

#### 3.3.1. Surface Morphology

The cross-sectional morphologies of FPGO/SPSU cross-linked PEM and Pt-TiO_2_/FPGO/SPSU-X nanocomposite PEMs are shown in Figure 4. It can be observed that evident aggregation and phase separation did not appear and Pt-TiO_2_ nanoparticles were uniformly distributed in the nanocomposite PEMs. If aggregation and phase separation occur, obvious fibrous and granular crystals appear on the section (as shown by red arrows in Figure 4d). This is because FPGO contribute to a stronger cross-linked interaction with SPSU compared with the pristine GO, which weakens the surface tension of GO and avoids micro-phase separation. When the content of Pt-TiO_2_ nanoparticles is 0.1%, the multilayer fold layer structure of the nanocomposite PEM can be clearly observed from the section diagram, indicating that FPGO and Pt-TiO_2_ nanoparticles have good compatibility with SPSU matrix. However, when the Pt-TiO_2_ content is increased up to 3 wt.%, some fibrous and granular crystals appeared on the cross section. The rough morphology and partial agglomeration of the inorganic additives is most likely caused by the excessive Pt nanoparticles affecting the dispersion of TiO_2_ and FPGO in SPSU.

#### 3.3.2. Mechanical Properties

As indicated in Table 1, the cross-linked network structure and incorporation of Pt-TiO_2_ nanoparticles have effect on the mechanical properties of all the membranes. When incorporating of Pt-TiO_2_ nanoparticles the nanocomposite PEMs, the elastic deformation resistance can be effectively improved while the tensile strength decreased to a certain extent. This is because the incorporation of Pt-TiO_2_ nanoparticles dispersed between the SPSU matrix and the lamellar structure of GO hinder the movement of the polymer chain segment. In addition, the increase in Pt-TiO_2_ contents will cause the aggregation of the inorganic additives and structural defects of the nanocomposite PEMs, which make it more prone to stress concentration under stress and thereby decrease the strength of the nanocomposite PEMs.

#### 3.3.3. Thermal Stability

The thermal stability of the as-obtained membranes, which is important for the PEM lifetime, is determined by TGA and displayed in Figure 5. Both the FPGO/SPSU cross-linked PEM and the Pt-TiO_2_/FPGO/SPSU-X nanocomposite PEMs show similar decomposition processes with a three-step weight loss behavior. The first weight loss platform of the nanocomposite membrane appeared below 200 °C, which was mainly caused by the volatilization of adsorbed water, bound water and residual solvent in the membranes. At this stage, the weight loss of the nanocomposite membrane was significantly greater than that of the cross-linked membrane, indicating that the incorporation of Pt-TiO_2_ nanoparticles could greatly improve the water retention capacity of the nanocomposite membrane. The second weight loss platform is in the range of 200~400 °C, corresponding to the decomposition of sulfonic acid groups and several other functional groups in composite membrane. In this stage, the weight loss curve of the nanocomposite membrane did not change significantly, indicating that the incorporation of Pt-TiO_2_ nanoparticles can lead to an improvement in the thermal stability of the nanocomposite membrane to a certain extent. The third weight loss platform starting from 400 °C was mainly caused by the decomposition of polymer main chains. In contrast, Nafion begins chemical decomposition at about 300 °C, and major decomposition products in the range 355 °C–560 °C were HF, SiF_4_, carbonyl fluorides, and species exhibiting C–F stretching vibrations [48]. Therefore, the cross-linked PEM and the nanocomposite PEMs are sufficiently stable below 200 °C and can meet the application of PEMFC.

#### 3.3.4. Oxidation Stability

The morphologies of FPGO/SPSU cross-linked PEM and Pt-TiO_2_/FPGO/SPSU-X nanocomposite PEM immersed in Fenton reagent at 80 °C for 210 min are shown in Figure 6 and Table 2. By observing the color of reagents in the sample bottle and the morphological changes of the membrane samples, it can be seen that the nanocomposite PEMs show good oxidation resistance when the cross-linked PEM was deformed and dissolved to different degrees. However, with an increase in the content of Pt-TiO_2_ nanoparticles within the nanocomposite PEMs, the hydrogen bond between GO and SPSU matrix is weakened by a large number of Pt-TiO_2_ nanoparticles dispersed in the polymer matrix, which make the polar groups in SPSU more vulnerable to the attack of -OH and -OOH radicals in Fenton reagent, and finally resulting in the decreased oxidation stability of the nanocomposite PEMs.

### 3.4. The Performance of the Cross-Linked PEM and the Nanocomposite PEMs

#### 3.4.1. Water Uptake, Swelling Ratio and Ion Exchange Capacity

WU and IEC of all the membranes are summarized in Table 3. As seen in the table, WU and IEC of FPGO/SPSU cross-linked PEM and Pt-TiO_2_/FPGO/SPSU-X nanocomposite PEMs are higher than that of the primary SPSU, indicating that the introduction of GO polymer brushes and the incorporation of Pt-TiO_2_ nanoparticles helps to increase hydrophilicity and free volume property of the membranes. However, WU and IEC increase initially and decrease afterwards with an increase in Pt-TiO_2_ nanoparticles contents. This is because more Pt-TiO_2_ nanoparticles cover the sulfonic groups in the nanocomposite membrane, resulting in a decrease in the number of effective sulfonic groups. The relatively high water absorption is not necessarily a good characteristic, because it will produce mechanical stress during the fuel cell operation.

SR of all the membranes at an elevated temperature are also shown in Table 3. SR of the nanocomposite PEM at 30 °C slightly increases with the increase in Pt-TiO_2_ nanoparticles content. However, the SR of Pt-TiO_2_/FPGO/SPSU-3 membrane at 70 °C and 90 °C is smaller than that of Pt-TiO_2_/FPGO/SPSU-1 and Pt-TiO_2_/FPGO/SPSU-2 nanocomposite membrane.

#### 3.4.2. Proton Conductivity

Figure 7a presents the temperature-dependent proton conductivity of the primary SPSU membrane, the FPGO/SPSU membrane and the nanocomposite membranes. All membranes exhibit a pronounced increase in proton conductivities with increasing temperature from 30 °C to 90 °C because of the thermally activated characteristic of the proton transportation. Arrhenius plots of proton conductivity of all membranes are also depicted in Figure 7b. For a purely Grotthuss mechanism, the expected activation energy is in the range of 14.3 to 39.8 kJ/mol, which matches the experimentally determined range of 15.31 to 22.96 kJ/mol. It can also be seen that the Arrhenius plot of all membranes shows a positive temperature-conductivity dependence, revealing the thermal activity process of proton conductivity. The significantly improved proton conductivity with elevated temperature can be attributed to the combined effect of the following factor: the increase in temperature will promote the movement of macromolecular chains and the absorption of water molecules in membrane materials.

However, the incorporation of Pt-TiO_2_ nanoparticles will slightly decrease the proton conductivity of the nanocomposite membrane to a certain extent. At the same time, the proton conductivity of the nanocomposite membrane obviously reduces with an increase in the content of incorporated Pt-TiO_2_ nanoparticles. Although Pt-TiO_2_ nanoparticles can improve the water uptake and ion exchange capacity of the nanocomposite membranes to a certain extent, the uneven distribution of the nanoparticles may affect the continuity of the proton transport channel formed by sulfonate ion clusters. In addition, the incorporation of Pt-TiO_2_ nanoparticles will occupy more free volume of the SPSU matrix, which does not benefit the formation of proton transport channels.

Herein, the proton conductivity of PEMs is strongly dependent on their channel structures and water uptake [51]. However, due to the chemical and surface properties of GO polymer brush and Pt-TiO_2_ nanoparticles as additives, the proton conductivity mechanism of the nanocomposites PEMs is becoming complicated and necessary to further investigate. Figure 8 indicates the mechanism of promoting proton transport in the nanocomposite PEMs. The hydroxyl groups on the Pt-TiO_2_ surface can interact with water molecules through hydrogen bonds, which is helpful for the formation of hydrophilic channels, so that the proton transfer can be quickly carried out in the form of hydrated hydrogen ions through the diffusion mechanism. Moreover, the hydrogen bond interaction between Pt-TiO_2_ and GO polymer brush as well as cross-linked polymer matrix also provides proton transport channels.

Figure 9 indicates the operating time-dependent proton conductivity of the primary SPSU membrane, the cross-linked PEM and the nanocomposite PEMs at 60 °C and room humidity (RH) of 39%. The membrane holder was suddenly heated to 60 °C while still in contact with ambient air (39% RH) to induce dehydration, which was monitored by membrane conductivity measurements. Compared with the primary SPSU membrane, the cross-linked membrane and the nanocomposite membranes exhibit higher proton conductivity, indicating that the nanocomposite membrane has better water retention capacity under high temperature conditions. This is because the hydrogen bonds between the hydroxyl groups on the surface of the nanoparticles and the water molecules in the system inhibit the movement of water, thereby reduce the evaporation rate of water molecules.

#### 3.4.3. Methanol Permeability and Selectivity

In a direct methanol fuel cell (DMFC), if the methanol on the anode side penetrates the proton exchange membrane to the cathode side, it will not only cause fuel waste and catalyst poisoning, but also bring some problems such as a drop in open circuit voltage of the battery [52]. Therefore, low methanol permeability and a higher selectivity coefficient is an inevitable requirement corresponding to the PEM used in DMFC.

Table 4 shows the methanol permeability and selectivity coefficient of Nafion 117, the primary SPSU membrane, the cross-linked PEM and the nanocomposite PEMs. The results show that under the same measurement conditions, the methanol permeabilities of Nafion 117 and the SPSU membrane are much higher than those of the cross-linked PEM and the nanocomposite PEMs. The introducing of GO polymer brushes and incorporating of Pt-TiO_2_ nanoparticles occupy more free volume of the cross-linked PEM and the nanocomposite PEMs and inhibit the movement of polymeric segments, thereby limiting the size of ion clusters that can be used as methanol permeation channels. In addition, the methanol permeabilities of the nanocomposite PEMs increase slightly as the Pt-TiO_2_ contents increase. This is because the dispersion of particles in the membrane decreases with an increase in Pt-TiO_2_ contents.

The conductivity and methanol resistance of the membrane are two important parameters which are mutually restrictive in many cases during proton exchange. The selectivity is defined as the ratio of the proton conductivity to the methanol permeability, which intuitively reflects their relationship. It can be seen from Table 4 that the incorporation of an appropriate amount of FPGO and Pt-TiO_2_ can significantly improve the selectivity of the cross-linked PEM and the nanocomposite PEMs.

## 4. Conclusions

Based on introducing GO polymer brushes as an inorganic additive and incorporating Pt-TiO_2_ nanoparticles fixed on polymeric PEM by forming cross-linked network structure, novel cross-linked PEM and nanocomposite PEMs with high performance and self-humidifying were prepared to solve leakage of inorganic additives during use and compatibility problem with organic polymers. Although the leakage of inorganic additives was not measured using an actual fuel cell, there is an indication that the leakage of inorganic additives may not occur based on SEM images. The introduction of GO polymer brushes into the nanocomposite PEMs exhibits a synergistic effect with the incorporating of Pt-TiO_2_ nanoparticles overcoming the trade-off effects between proton conductivity and water retention ability, proton conductivity and methanol permeability. Proton conductivity, water uptake and methanol permeability of the nanocomposite PEM can be up to 6.93 mS cm^−1^, 46.58% and be as low as 1.4157 × 10^−6^ cm^2^ s^−1^, respectively, which represent an increase of about 70%, about 22% and a decrease of about 40%, respectively, compared with that of primary SPSU. The study will provide a new method toward achieving high proton conductivity and water uptake as well as low methanol permeability, which has a good application prospect for direct methanol fuel cell.

## Figures and Tables

**Figure 1 nanomaterials-12-00841-f001:**
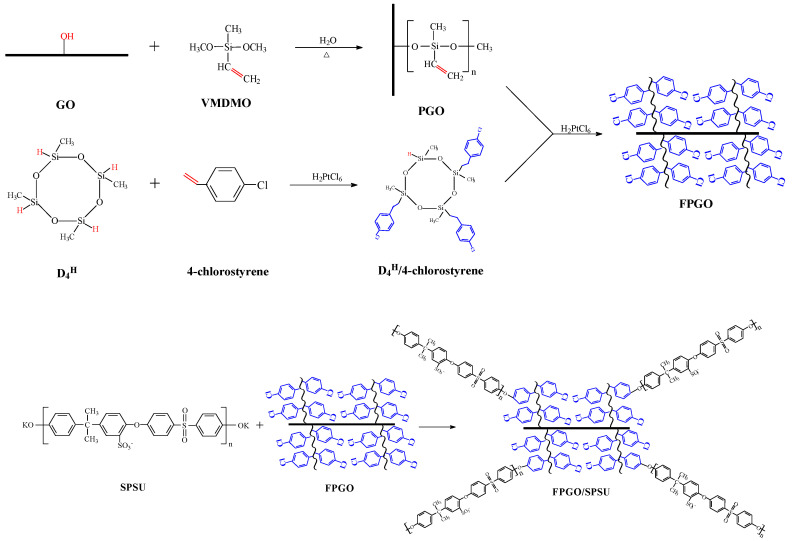
Schematic representation of preparation process of FPGO and FPGO/SPSU cross-linked PEM.

**Figure 2 nanomaterials-12-00841-f002:**
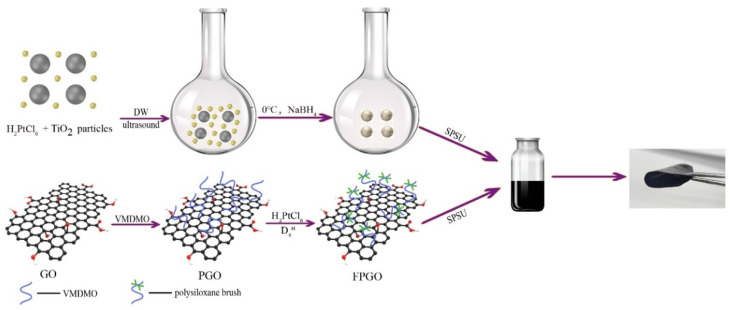
Schematic illustration of the fabrication process for Pt-TiO_2_/FPGO/SPSU nanocomposite PEM.

**Figure 3 nanomaterials-12-00841-f003:**
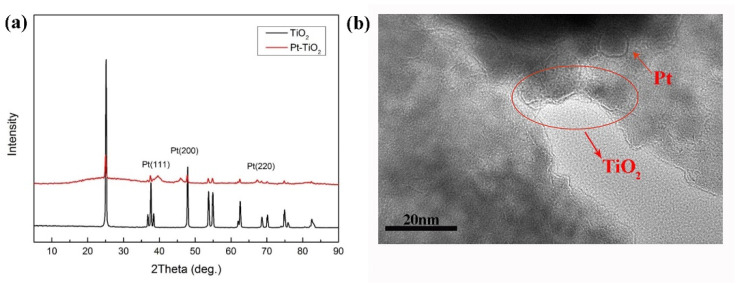
X-ray diffraction pattern (**a**) and TEM image (**b**) of Pt-TiO_2_ nanoparticles.

**Figure 4 nanomaterials-12-00841-f004:**
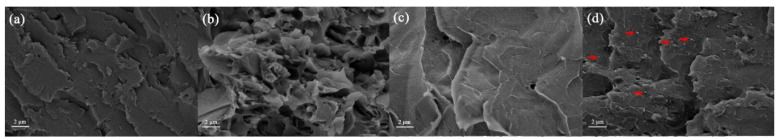
SEM images of the membranes: (**a**) FPGO/SPSU, (**b**) Pt-TiO_2_/FPGO/SPSU-1, (**c**) Pt-TiO_2_/FPGO/SPSU-2, (**d**) Pt-TiO_2_/FPGO/SPSU@-3.

**Figure 5 nanomaterials-12-00841-f005:**
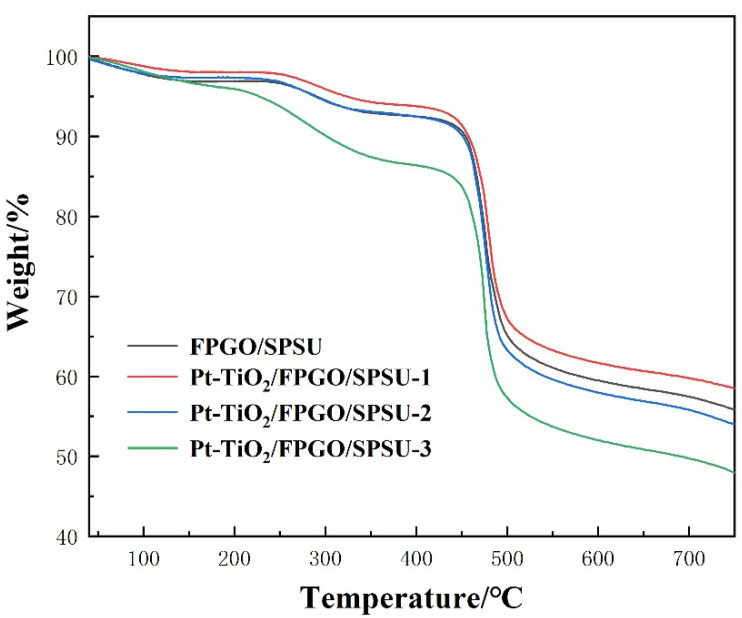
TGA curves of all the membranes.

**Figure 6 nanomaterials-12-00841-f006:**
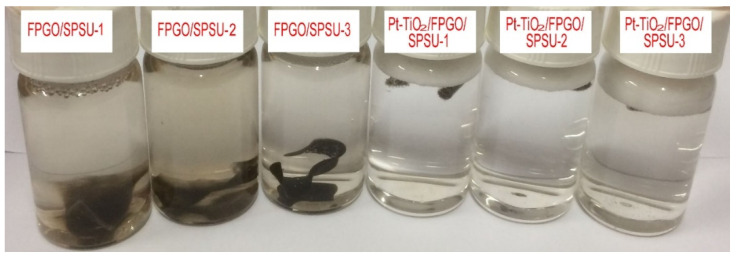
Photograph of all membranes immersed in Fenton reagent for 210 min at 80 °C.

**Figure 7 nanomaterials-12-00841-f007:**
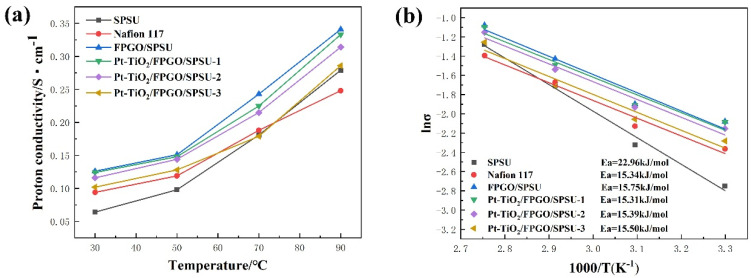
The temperature-dependent proton conductivity (**a**) and Arrhenius plots of proton conductivity (**b**) for all the membranes. (The data of Nafion 117 in Figure 7 are from ref. [44].)

**Figure 8 nanomaterials-12-00841-f008:**
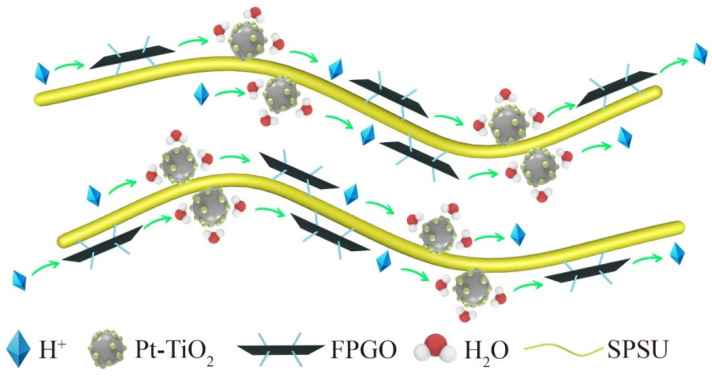
Diagram of proton conductivity mechanism in the nanocomposite PEM.

**Figure 9 nanomaterials-12-00841-f009:**
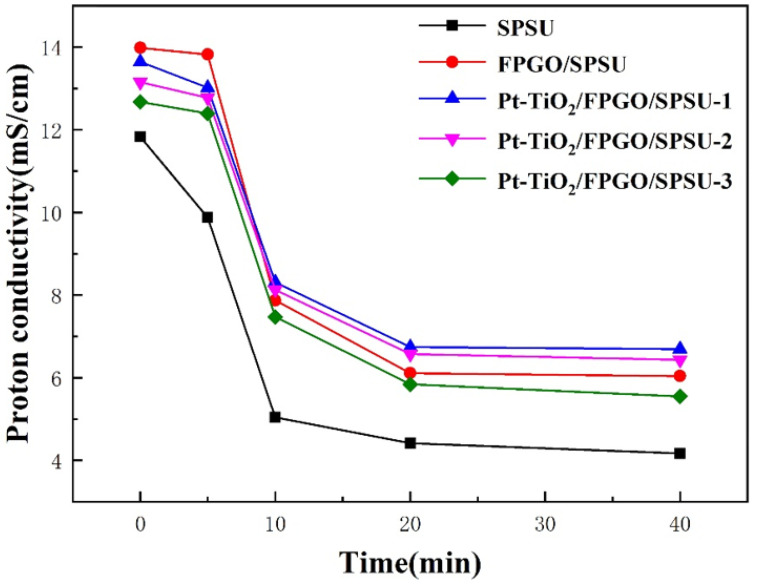
Proton conductivity of all the membranes at a constant temperature of 60 °C with time.

**Table 1 nanomaterials-12-00841-t001:** Mechanical properties of all the membranes.

PEM Samples	Young’s Modulus (MPa)	Tensile Strength (MPa)	Elongation at Break (%)
Nafion 117	100.00 ^a^	28.40 ^a^	329 ^a^
SPSU	342.30	30.15	7.29
FPGO/SPSU	267.50	32.60	12.37
Pt-TiO_2_/FPGO/SPSU-1	304.50	29.14	15.44
Pt-TiO_2_/FPGO/SPSU-2	671.10	30.90	13.81
Pt-TiO_2_/FPGO/SPSU-3	258.05	25.44	12.82

^a^ According to ref. [47].

**Table 2 nanomaterials-12-00841-t002:** Oxidative stability of all the membranes.

PEM Samples	Oxidative Stability ^a^ (min)		
	τ_1_ ^b^	τ_2_ ^c^	Δ = τ_2_ − τ_1_
Nafion 117	180 ^d^	>960 ^d^	>780 ^d^
SPSU	75	120	45
FPGO/SPSU	90	310	220
Pt-TiO_2_/FPGO/SPSU-1	255	735	480
Pt-TiO_2_/FPGO/SPSU-2	270	630	360
Pt-TiO_2_/FPGO/SPSU-3	240	545	305

^a^ Measured at 80 °C in 3% H_2_O_2_ containing 2 ppm FeSO_4_. ^b^ The time when the membrane began to dissolve. ^c^ The time when the membrane dissolved completely. ^d^ According to ref. [49].

**Table 3 nanomaterials-12-00841-t003:** WU, IEC and SR of all the membranes.

Membrane Samples	WU (%)	IEC (mmol/g)	SR (%)			
			30 °C	50 °C	70 °C	90 °C
Nafion 117	35.60 ^a^	0.91 ^a^	13.02 ^b^	15.88 ^b^	17.52 ^b^	20.16 ^b^
SPSU	38.12	1.52	6.02	13.73	20.42	169.72
FPGO/SPSU	45.89	1.83	8.85	12.62	14.93	154.39
Pt-TiO_2_/FPGO/SPSU-1	46.58	1.91	9.16	13.37	15.87	172.52
Pt-TiO_2_/FPGO/SPSU-2	47.85	2.23	9.21	14.25	16.74	187.36
Pt-TiO_2_/FPGO/SPSU-3	47.81	1.97	9.47	14.76	15.31	166.27

^a^ According to ref. [47]. ^b^ According to ref. [50].

**Table 4 nanomaterials-12-00841-t004:** Methanol permeability and selectivity of Nafion 117 and all the membranes.

Membrane Samples	Methanol Permeability (10^−6^ cm^2^ s^−1^)	Selectivity (10^4^ S s cm^−3^)
Nafion 117	2.9400 ^a^	4.2619
SPSU	2.3407	4.7934
FPGO/SPSU	1.7117	7.2267
Pt-TiO_2_/FPGO/SPSU-1	1.4157	8.5117
Pt-TiO_2_/FPGO/SPSU-2	1.6186	7.1667
Pt-TiO_2_/FPGO/SPSU-3	1.6853	6.0879

^a^ According to ref. [53].

## Data Availability

Not applicable.

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
