# Peer review of "High Performance and Self-Humidifying of Novel Cross-Linked and Nanocomposite Proton Exchange Membranes Based on Sulfonated Polysulfone"

_nanomaterials, 2022, doi:10.3390/nano12050841_

Round 1
Reviewer 1 Report
Dear Professor Zhang,
The review of manuscript with the reference number Nanomaterials-1583312 has been conducted carefully, and the concerns have been enclosed as following:
- The English and academic level of writing should be improved.
- the introduction of manuscript needs to be improved by literature pertaining to fluorine-based PEMs, and then the drawbacks should be highlighted. It comes from the tremendous use of such materials. For example, take a look at these contribution for more justification: https://doi.org/10.1016/j.energy.2021.121940, https://doi.org/10.1016/j.ssi.2020.115343.
- The procedure of nanomaterial release test should be included. There is no data to show the leakage of nanoparticles.
- 9: the performance of the membranes in prolonged time will be prolific to discuss the durability of the performance.
Author Response
We would like to thank the editor for giving us a chance to resubmit the paper, and also thank the reviewers for giving us constructive suggestions which would help us both in English and in depth to improve the quality of the paper. Here we submit a new version of our manuscript (nanomaterials-1583312), which has been modified according to your suggestions and the reviewers’ suggestions. Efforts were also made to correct the mistakes and improve the English of the manuscript. The revised words and sentences were marked with red in the revised manuscript.
The review of manuscript with the reference number Nanomaterials-1583312 has been conducted carefully, and the concerns have been enclosed as following:
(1) The English and academic level of writing should be improved
Answer: Thank the reviewer for the comments very much. The reviewer’s suggestions have been adopted. We have revised the WHOLE manuscript carefully and tried to avoid any grammar or syntax error. In addition, we have asked several colleagues who are skilled authors of English language papers to check the English.
(2) the introduction of manuscript needs to be improved by literature pertaining to fluorine-based PEMs, and then the drawbacks should be highlighted. It comes from the tremendous use of such materials. For example, take a look at these contribution for more justification: https://doi.org/10.1016/j.energy.2021.121940, https://doi.org/10.1016/j.ssi.2020.115343.
Answer: The drawbacks of fluorine-based PEMs have been added into the introduction section in the revised manuscript. And the relevant data for Nafion 117 have been added to Figures 6, 7 and Tables 1, 2, 3 for comparison with this work.
(3) The procedure of nanomaterial release test should be included. There is no data to show the leakage of nanoparticles.
Answer: There may be no leakage of inorganic additives because Pt-TiO2 nanoparticles were not observed according to the cross-section SEM images of the membrane samples in the paper.
(4) 9: the performance of the membranes in prolonged time will be prolific to discuss the durability of the performance.
Answer: The measurement for a long time in Figure 9 has not been carried out because obvious differences for the self-humidifying performance have been obtained.
Reviewer 2 Report
nanomaterials-1583312
High performance and self-humidifying of novel cross-linked and nanocomposite proton exchange membranes based on sulfonated polysulfone
Xinyu Li, Zhongxin Zhang, Xinrui Guo, Tianjian Yang, Zhongli Li, Huaxin Rao
Summary
The proposed membrane composition appears to be new. However, the contribution is difficult to evaluate because results are not compared to an existing and prevalent material such as Nafion. Furthermore, the discussion could be enhanced and supported by additional references, and many suggestions are provided. For these reasons, the manuscript is not yet ready to be accepted.
Comments
- Nafion or other baseline data should be added to most figures (5 to 7, and 9) and tables (1 to 3). Otherwise, it is not possible to appreciate the real value of this contribution.
- The English language needs a revision.
- Section 2.5.3, 1st It is stated that “Then the membrane samples were immersed in ionic water for 48 h, dried in an oven at 30°C for 24 h, and removed. The water on the membrane surface was wiped off with filter paper, and the mass of the membrane samples were weighed as Wwet (g).”. Why is the membrane dried after wetting it to obtain the wet weight? These 2 sentences should be revised.
- Section 2.5.4. It is expected that the membrane sample is first exposed to an acid solution to ensure it is in an acid form. After a rinse, the sample is exposed to a salt solution to exchange the protons and that solution is titrated to obtain the ion exchange capacity. Is that correct? The entire procedure should be reported.
- Section 2.5.9. Revise the title of this section as it is the same as section 2.5.8.
- Equation 5. What is the meaning of S?
- Figure 2. Clarify the purpose of the added Pt particles, which are not mentioned in section 2.3.
- Figure 3. Clarify the features identified by a red arrow and ellipse in figure 3b. Derive the size of the Pt and TiO2 particles using the Scherrer equation and figure 3a data for comparison with other measurements.
- Section 3.3.1. It is stated that “It can be observed that evident aggregation and phase separation did not appear and Pt-TiO2 nanoparticles were uniformly distributed in the nanocomposite PEMs.”. What is expected if aggregation and phase separation occur? This information should be added to the text. It is also stated that “The rough morphology and partial agglomeration of the inorganic additives is most likely caused by the excessive Pt nanoparticles affecting the dispersion of TiO2 and FPGO in SPSU.”. This sentence is unclear considering that the amount of Pt loaded on TiO2 was not varied.
- Section 3.3.3. It is stated that “The second weight loss platform is in the range of 200~400°C, corresponding to the decomposition of sulfonic acid groups and oxygen-containing functional groups in GO.”. The composite polymer contains several functional groups (figures 1 and 2). How can the second decomposition be attributed to only sulfonic acid and oxygen groups?
- Section 3.3.4, last sentence. Should “decreased” be used rather than “increased”?
- Table 2, note b. How is τ1 identified?
- Section 3.4.1, 1st It is stated that “This is because more Pt-TiO2 nanoparticles cover the sulfonic groups in the nanocomposite membrane, resulting in replacing the number of effective sulfonic groups with decreasing of Na ion.”. The meaning of Na ion in this context is unclear as a proton form is the target for fuel cells. The water uptake is relatively high and is not necessarily a good feature because it would create mechanical stresses during fuel cell operation. A comment should be added to emphasize that point.
- Section 3.4.1, 2nd It is stated that “The decrease is caused by the fact that polar interaction and dynamic cross-linking between ‒SO3H and TiO2 nanoparticles on the SPSU matrix limit the further swelling of the nanocomposite membrane.”. What evidence can be given to draw this conclusion?
- Section 3.4.2, 1st It is stated that “The Ea values of all membranes which are between 15.31~22.96 kJ/mol by the Arrhenius plot are in the range of 14.3~39.8 kJ/mol by explaining the proton penetration behavior by Grotthuss mechanism.”. Clarify the meaning of this sentence. For instance, are the authors suggesting that for a purely Grotthuss mechanism the expected activation energy is in the range of 14.3 to 39.8 kJ/mol, which matches the experimentally determined range of 15.31 to 22.96 kJ/mol? Also, it is stated that “The significantly improved proton conductivity with elevated temperature can be attributed to the combined effect of the following two factors: (1) the increase of temperature will promote the movement of macromolecular chains and the absorption of water molecules in membrane materials, and (2) the increase of proton concentration due to the decrease of water dilution effect.”. However, the 2 listed explanations are seemingly contradictory. How can the water dilution effect be less important if more water is absorbed in the polymer? This contradiction needs to be resolved.
- Figure 9. What is the original relative humidity or water content of the membrane? Conditions were clearly changed to create membrane dehydration. However, the method used is not described.
- Section 3.4.3, 2nd It is stated that “Finally, the compactness of the product with high Pt-TiO2 content leads to faster methanol penetration.”. The meaning of that sentence is unclear in part because the density of these membranes was not measured.
- It is stated that “Based on introducing GO polymer brushes as an inorganic additive and incorporating Pt-TiO2 nanoparticles fixed on polymeric PEM by forming cross-linked network structure, novel cross-linked PEM and nanocomposite PEMs with high performance and self-humidifying were prepared to solve leakage of inorganic additives during use and compatibility problem with organic polymers.”. The authors cannot comment on the leakage of inorganic additives because this aspect was not quantitatively evaluated. Although GO was cross-linked, the Pt-TiO2 particles are only maintained by weak hydrogen bonds.
Author Response
We would like to thank the editor for giving us a chance to resubmit the paper, and also thank the reviewers for giving us constructive suggestions which would help us both in English and in depth to improve the quality of the paper. Here we submit a new version of our manuscript (nanomaterials-1583312), which has been modified according to your suggestions and the reviewers’ suggestions. Efforts were also made to correct the mistakes and improve the English of the manuscript. The revised words and sentences were marked with red in the revised manuscript.
The proposed membrane composition appears to be new. However, the contribution is difficult to evaluate because results are not compared to an existing and prevalent material such as Nafion. Furthermore, the discussion could be enhanced and supported by additional references, and many suggestions are provided. For these reasons, the manuscript is not yet ready to be accepted.
Comments
(1) Nafion or other baseline data should be added to most figures (5 to 7, and 9) and tables (1 to 3). Otherwise, it is not possible to appreciate the real value of this contribution.
Answer: The relevant data of Nafion 117 have been supplemented into Figure 6, 7 and Tables 1, 2 and 3.
(2) The English language needs a revision.
Answer: Thank the reviewer very much for indicating the error in our manuscript. We have revised the whole manuscript carefully and tried to avoid any grammar or syntax error. In addition, we have asked several colleagues who are skilled authors of English language papers to check the English.
(3) Section 2.5.3, 1st It is stated that “Then the membrane samples were immersed in ionic water for 48 h, dried in an oven at 30°C for 24 h, and removed. The water on the membrane surface was wiped off with filter paper, and the mass of the membrane samples were weighed as Wwet (g).”. Why is the membrane dried after wetting it to obtain the wet weight? These 2 sentences should be revised.
Answer: Two incorrect statements in Section 2.5.3 have been modified as “Then the membrane samples were immersed in ionic water for 48 h and removed, the water on the membrane surface was wiped off with filter paper, and the mass of the membrane samples were weighed as Wwet ( g )”.
(4) Section 2.5.4. It is expected that the membrane sample is first exposed to an acid solution to ensure it is in an acid form. After a rinse, the sample is exposed to a salt solution to exchange the protons and that solution is titrated to obtain the ion exchange capacity. Is that correct? The entire procedure should be reported.
Answer: The IEC test process was supplemented by “Ion exchange capacity ( IEC ) of the membrane sample was determined by acid-base titration. Firstly, membrane samples were soaked in 1 M HCl solution and heated to 70°C for 1 h, which changed them into the H+ form. The samples were then washed with distilled water several times to remove excess HCl, right after that soaked them in boiling water for 1 h which confirmed the stability of membranes in hydrolytic condition. Next step, we take the samples in 50 ml of 1 M NaCl solution heated to 40°C and equilibrated for at least 24 h to replace the protons by sodium ions. The remaining solution was titrated with 0.01 M NaOH solution using phenolphthalein as an indicator.”
(5) Section 2.5.9. Revise the title of this section as it is the same as section 2.5.8.
Answer: The title of this section is revised as “Measurement of methanol permeability and selectivity”.
(6) Equation 5. What is the meaning of S?
Answer: S is the slope of the time-dependent molar concentration of methanol solution in room B.
(7) Figure 2. Clarify the purpose of the added Pt particles, which are not mentioned in section 2.3.
Answer: Pt particles exhibit hydrophilic property and can adsorb water molecules. Hence, Pt particles are used to modify TiO2 to improve further the self-humidifying performance of PEMs.
(8) Figure 3. Clarify the features identified by a red arrow and ellipse in figure 3b. Derive the size of the Pt and TiO2 particles using the Scherrer equation and figure 3a data for comparison with other measurements.
Answer: The red arrow and ellipse in figure 3b indicate that Pt nanoparticles are deposited on TiO2. The size of Pt nanoparticles deposited on TiO2 particles calculated by Scherrer equation is about 2.05 nm.
(9) Section 3.3.1. It is stated that “It can be observed that evident aggregation and phase separation did not appear and Pt-TiO2 nanoparticles were uniformly distributed in the nanocomposite PEMs.”. What is expected if aggregation and phase separation occur? This information should be added to the text. It is also stated that “The rough morphology and partial agglomeration of the inorganic additives is most likely caused by the excessive Pt nanoparticles affecting the dispersion of TiO2 and FPGO in SPSU.”. This sentence is unclear considering that the amount of Pt loaded on TiO2 was not varied.
Answer: If aggregation and phase separation occur, obvious fibrous and granular crystals appear on the section.
To avoid partial agglomeration of the inorganic additives, proper Pt was used to form Pt-TiO2 nanoparticles. However, when the Pt-TiO2 content is increased up to 3 wt.%, some fibrous and granular crystals appeared on the cross section.
(10) Section 3.3.3. It is stated that “The second weight loss platform is in the range of 200~400°C, corresponding to the decomposition of sulfonic acid groups and oxygen-containing functional groups in GO.”. The composite polymer contains several functional groups (figures 1 and 2). How can the second decomposition be attributed to only sulfonic acid and oxygen groups?
Answer: In Section 3.3.3,“The second weight loss platform is in the range of 200~400°C, corresponding to the decomposition of sulfonic acid groups and oxygen-containing functional groups in GO.” has been modified as “The second weight loss platform is in the range of 200~400°C, corresponding to the decomposition of sulfonic acid groups and other functional groups in composite membrane.”
(11) Section 3.3.4, last sentence. Should “decreased” be used rather than “increased”?
Answer: In the last sentence of section 3.3.4, “increased” had been revised as “decreased”.
(12) Table 2, note b. How is τ1 identified?
Answer: The quality of the membrane sample was weighed before adding Fenton reagent, and then the quality of the membrane sample was weighed every five minutes until the quality of the membrane sample changed. In the paper, τ1 and τ2 mean the time when the membrane began to dissolve and the time when the membrane dissolved completely, respectively.
(13) Section 3.4.1, 1st It is stated that “This is because more Pt-TiO2 nanoparticles cover the sulfonic groups in the nanocomposite membrane, resulting in replacing the number of effective sulfonic groups with decreasing of Naion.”. The meaning of Naion in this context is unclear as a proton form is the target for fuel cells. The water uptake is relatively high and is not necessarily a good feature because it would create mechanical stresses during fuel cell operation. A comment should be added to emphasize that point.
Answer: Section 3.4.1, the sentence “This is because more Pt-TiO2 nanoparticles cover the sulfonic groups in the nanocomposite membrane, resulting in replacing the number of effective sulfonic groups with decreasing of Naion.” has been modified as “This is because more Pt-TiO2 nanoparticles cover the sulfonic groups in the nanocomposite membrane, resulting in a decrease in the number of effective sulfonic groups.” In addition, the sentence “The relatively high water absorption is not necessarily a good characteristic, because it will produce mechanical stress during the fuel cell operation” is added into the revised manuscript to clarify the influence of excessive water.
(14) Section 3.4.1, 2nd It is stated that “The decrease is caused by the fact that polar interaction and dynamic cross-linking between ‒SO3H and TiO2 nanoparticles on the SPSU matrix limit the further swelling of the nanocomposite membrane.”. What evidence can be given to draw this conclusion?
Answer: The conclusion in Section 3.4.1 “Finally, the compactness of the product with high Pt-TiO2 content leads to faster methanol penetration”, has been removed in the revised manuscript to avoid some misleading.
(15) Section 3.4.2, 1st It is stated that “The Ea values of all membranes which are between 15.31~22.96 kJ/mol by the Arrhenius plot are in the range of 14.3~39.8 kJ/mol by explaining the proton penetration behavior by Grotthuss mechanism.”. Clarify the meaning of this sentence. For instance, are the authors suggesting that for a purely Grotthuss mechanism the expected activation energy is in the range of 14.3 to 39.8 kJ/mol, which matches the experimentally determined range of 15.31 to 22.96 kJ/mol? Also, it is stated that “The significantly improved proton conductivity with elevated temperature can be attributed to the combined effect of the following two factors: (1) the increase of temperature will promote the movement of macromolecular chains and the absorption of water molecules in membrane materials, and (2) the increase of proton concentration due to the decrease of water dilution effect.”. However, the 2 listed explanations are seemingly contradictory. How can the water dilution effect be less important if more water is absorbed in the polymer? This contradiction needs to be resolved.
Answer: Section 3.4.2, the sentence “The Ea values of all membranes which are between 15.31~22.96 kJ/mol by the Arrhenius plot are in the range of 14.3~39.8 kJ/mol by explaining the proton penetration behavior by Grotthuss mechanism.” has been modified as “For a purely Grotthuss mechanism the expected activation energy is in the range of 14.3 to 39.8 kJ/mol, which matches the experimentally determined range of 15.31 to 22.96 kJ/mol.” In addition, the sentence “the increase of proton concentration due to the decrease of water dilution effect.” has been deleted in the revised manuscript to avoid some misleading.
(16) Figure 9. What is the original relative humidity or water content of the membrane? Conditions were clearly changed to create membrane dehydration. However, the method used is not described.
Answer: The whole test mould was placed in a thermostat at 60°C. The longer the test mould was in the thermostat, the less the water content in the membrane sample was. When the membrane sample reaches room temperature and 39% RH, the corresponding proton conductivity can be calculated according to formula (4) by measuring the impedance value of the membrane samples for every 5 min.
(17) Section 3.4.3, 2nd It is stated that “Finally, the compactness of the product with high Pt-TiO2 content leads to faster methanol penetration.”. The meaning of that sentence is unclear in part because the density of these membranes was not measured.
Answer: The sentence “Finally, the compactness of the product with high Pt-TiO2 content leads to faster methanol penetration”, has been removed in the revised manuscript to avoid some misleading because the density of these membranes was not measured.
(18) It is stated that “Based on introducing GO polymer brushes as an inorganic additive and incorporating Pt-TiO2 nanoparticles fixed on polymeric PEM by forming cross-linked network structure, novel cross-linked PEM and nanocomposite PEMs with high performance and self-humidifying were prepared to solve leakage of inorganic additives during use and compatibility problem with organic polymers.”. The authors cannot comment on the leakage of inorganic additives because this aspect was not quantitatively evaluated. Although GO was cross-linked, the Pt-TiO2 particles are only maintained by weak hydrogen bonds.
Answer: There may be no leakage of inorganic additives because Pt-TiO2 nanoparticles were not observed according to the cross-section SEM images of the membrane samples in the paper.
Reviewer 3 Report
Comments: Major revision
- In the title of section 2.5.9: it seems that “Measurement of proton conductivity” should be “2.5.9. Measurement of methanol permeability”
- In figure 3b, the TEM image of Pt-TiO2 nanoparticles is not clear.
- The curves of the tensile test should be provided.
- The quality of all figures should be improved.
- Fuel cell test of membranes should be provided.
- There are high grammatical errors in the manuscript.
- The performance of membranes should be compared with other similar works.
Author Response
We would like to thank the editor for giving us a chance to resubmit the paper, and also thank the reviewers for giving us constructive suggestions which would help us both in English and in depth to improve the quality of the paper. Here we submit a new version of our manuscript (nanomaterials-1583312), which has been modified according to your suggestions and the reviewers’ suggestions. Efforts were also made to correct the mistakes and improve the English of the manuscript. The revised words and sentences were marked with red in the revised manuscript.
(1) In the title of section 2.5.9: it seems that “Measurement of proton conductivity” should be “2.5.9. Measurement of methanol permeability”
Answer: The title of the section 2.5.9 is revised as “Measurement of methanol permeability and selectivity”.
(2) In figure 3b, the TEM image of Pt-TiO2 nanoparticles is not clear.
Answer: The TEM image of Pt-TiO2 nanoparticles has been revised.
(3) The curves of the tensile test should be provided.
Answer: According to other similar literatures, some key data of the mechanical properties were also showed through the table form, so there was no supplement to the tensile curve in the revised manuscript.
(4) The quality of all figures should be improved.
Answer: The quality of some figures has been revised.
(5) Fuel cell test of membranes should be provided.
Answer: The research contents in the paper deal mainly with performance of the nanocomposite PEMs. In future study, the nanocomposite PEMs will be assembled to fuel cell and their performance will be further investigated.
(6) There are high grammatical errors in the manuscript.
Answer: We have revised the whole manuscript carefully and tried to avoid any grammar or syntax error. In addition, we have asked several colleagues who are skilled authors of English language papers to check the English.
(7) The performance of membranes should be compared with other similar works.
Answer: The relevant data for Nafion 117 have been added to Figures 6, 7 and Tables 1, 2, 3 for comparison with this work.
Round 2
Reviewer 2 Report
Summary
Most comments were considered. However, several clarifications are still needed.
Comments
- Nafion or other baseline data should be added to most figures (5 to 7, and 9) and tables (1 to 3). Otherwise, it is not possible to appreciate the real value of this contribution.
- Answer: The relevant data of Nafion 117 have been supplemented into Figure 6, 7 and Tables 1, 2 and 3.
- Figure 5 (section 3.3.3) was not supplemented with a discussion of Nafion data (for example, Polymer, 39 (1998) 5961). Nafion data was not added to figure 6 although it is mentioned that the figure was updated. Add the source of the Nafion data in figure 7.
- The English language needs a revision.
- Answer: Thank the reviewer very much for indicating the error in our manuscript. We have revised the whole manuscript carefully and tried to avoid any grammar or syntax error. In addition, we have asked several colleagues who are skilled authors of English language papers to check the English.
- The English language still needs a revision. For instance, “ionic water” should likely be replaced by “deionized water” (line 148). Replace “room” by “compartment” (line 205). Correct “as shown in the red arrow logo in Figure 4d” by “as shown by red arrows in Figure 4d” (lines 258-259).
- Figure 2. Clarify the purpose of the added Pt particles, which are not mentioned in section 2.3.
- Answer: Pt particles exhibit hydrophilic property and can adsorb water molecules. Hence, Pt particles are used to modify TiO2 to improve further the self-humidifying performance of PEMs.
- Section 2.3 indicates that Pt is added to the mixture as chloroplatinic acid and not as Pt particles. However, figure 2 indicates that Pt is added as particles and chloroplatinic acid. This inconsistency should be resolved.
- Figure 3. Clarify the features identified by a red arrow and ellipse in figure 3b. Derive the size of the Pt and TiO2 particles using the Scherrer equation and figure 3a data for comparison with other measurements.
- Answer: The red arrow and ellipse in figure 3b indicate that Pt nanoparticles are deposited on TiO2. The size of Pt nanoparticles deposited on TiO2 particles calculated by Scherrer equation is about 2.05 nm.
- Add labels to figure 3b ellipse (TiO2 particle?) and arrow (Pt nanoparticle?). Add a brief explanation to the text considering that the Pt particle size obtained by XRD (2.05 nm) is different than the value derived from TEM (~10 nm).
- Section 3.3.1. It is stated that “It can be observed that evident aggregation and phase separation did not appear and Pt-TiO2 nanoparticles were uniformly distributed in the nanocomposite PEMs.”. What is expected if aggregation and phase separation occur? This information should be added to the text. It is also stated that “The rough morphology and partial agglomeration of the inorganic additives is most likely caused by the excessive Pt nanoparticles affecting the dispersion of TiO2 and FPGO in SPSU.”. This sentence is unclear considering that the amount of Pt loaded on TiO2 was not varied.
- Answer: If aggregation and phase separation occur, obvious fibrous and granular crystals appear on the section. To avoid partial agglomeration of the inorganic additives, proper Pt was used to form Pt-TiO2 However, when the Pt-TiO2 content is increased up to 3 wt.%, some fibrous and granular crystals appeared on the cross section.
- In section 2.4, second paragraph, indicate the weight ratio percentage of Pt-TiO2 nanoparticles to FPGO/SPSU corresponding to the 2.0 mg of Pt-TiO2
- Section 3.4.2, 1st It is stated that “The Ea values of all membranes which are between 15.31~22.96 kJ/mol by the Arrhenius plot are in the range of 14.3~39.8 kJ/mol by explaining the proton penetration behavior by Grotthuss mechanism.”. Clarify the meaning of this sentence. For instance, are the authors suggesting that for a purely Grotthuss mechanism the expected activation energy is in the range of 14.3 to 39.8 kJ/mol, which matches the experimentally determined range of 15.31 to 22.96 kJ/mol? Also, it is stated that “The significantly improved proton conductivity with elevated temperature can be attributed to the combined effect of the following two factors: (1) the increase of temperature will promote the movement of macromolecular chains and the absorption of water molecules in membrane materials, and (2) the increase of proton concentration due to the decrease of water dilution effect.”. However, the 2 listed explanations are seemingly contradictory. How can the water dilution effect be less important if more water is absorbed in the polymer? This contradiction needs to be resolved.
- Answer: Section 3.4.2, the sentence “The Ea values of all membranes which are between 15.31~22.96 kJ/mol by the Arrhenius plot are in the range of 14.3~39.8 kJ/mol by explaining the proton penetration behavior by Grotthuss mechanism.” has been modified as “For a purely Grotthuss mechanism the expected activation energy is in the range of 14.3 to 39.8 kJ/mol, which matches the experimentally determined range of 15.31 to 22.96 kJ/mol.” In addition, the sentence “the increase of proton concentration due to the decrease of water dilution effect.” has been deleted in the revised manuscript to avoid some misleading.
- The suggested change related to the Grotthuss mechanism was not implemented in the revised manuscript.
- Figure 9. What is the original relative humidity or water content of the membrane? Conditions were clearly changed to create membrane dehydration. However, the method used is not described.
- Answer: The whole test mould was placed in a thermostat at 60°C. The longer the test mould was in the thermostat, the less the water content in the membrane sample was. When the membrane sample reaches room temperature and 39% RH, the corresponding proton conductivity can be calculated according to formula (4) by measuring the impedance value of the membrane samples for every 5 min.
- This explanation is still unclear. It might be much clearer to only say that the membrane holder was suddenly heated to 60 °C while still in contact with ambient air (39 % RH) to induce dehydration, which was monitored by membrane conductivity measurements, assuming this is correct.
- It is stated that “Based on introducing GO polymer brushes as an inorganic additive and incorporating Pt-TiO2 nanoparticles fixed on polymeric PEM by forming cross-linked network structure, novel cross-linked PEM and nanocomposite PEMs with high performance and self-humidifying were prepared to solve leakage of inorganic additives during use and compatibility problem with organic polymers.”. The authors cannot comment on the leakage of inorganic additives because this aspect was not quantitatively evaluated. Although GO was cross-linked, the Pt-TiO2 particles are only maintained by weak hydrogen bonds.
- Answer: There may be no leakage of inorganic additives because Pt-TiO2 nanoparticles were not observed according to the cross-section SEM images of the membrane samples in the paper.
- This statement is still misleading because the leakage of inorganic additives was not measured using an actual fuel cell. It would be preferable to mention this clearly and add that there is an indication that the leakage of inorganic additives may not occur based on SEM images.
Author Response
The following is a point-to-point response to the note and the reviewers’ comments.
- Response to the Reviewer
- Nafion or other baseline data should be added to most figures (5 to 7, and 9) and tables (1 to 3). Otherwise, it is not possible to appreciate the real value of this contribution.
Answer: The relevant data of Nafion 117 have been supplemented into Figure 6, 7 and Tables 1, 2 and 3.
Figure 5 (section 3.3.3) was not supplemented with a discussion of Nafion data (for example, Polymer, 39 (1998) 5961). Nafion data was not added to figure 6 although it is mentioned that the figure was updated. Add the source of the Nafion data in figure 7.
Answer: In section 3.3.3, the sentence “In contrast, Nafion begins chemical decomposition at about 300°C, major decomposition products in the range 355°C–560°C were HF, SiF4, carbonyl fluorides, and species exhibiting C–F stretching vibrations.” is added into the revised manuscript to supplement with a discussion of Nafion data. In addition, the source of Nafion data for Figure 7 is from Reference 44 and added into the revised manuscript.
- The English language needs a revision.
Answer: Thank the reviewer very much for indicating the error in our manuscript. We have revised the whole manuscript carefully and tried to avoid any grammar or syntax error. In addition, we have asked several colleagues who are skilled authors of English language papers to check the English.
The English language still needs a revision. For instance, “ionic water” should likely be replaced by “deionized water” (line 148). Replace “room” by “compartment” (line 205). Correct “as shown in the red arrow logo in Figure 4d” by “as shown by red arrows in Figure 4d” (lines 258-259).
Answer: The words “ionic water”, “room” and “logo” had been modified and deleted in the revised manuscript. In addition, we have carefully checked the manuscript to avoid some mistakes.
- Figure 2. Clarify the purpose of the added Pt particles, which are not mentioned in section 2.3.
Answer: Pt particles exhibit hydrophilic property and can adsorb water molecules. Hence, Pt particles are used to modify TiO2 to improve further the self-humidifying performance of PEMs.
Section 2.3 indicates that Pt is added to the mixture as chloroplatinic acid and not as Pt particles. However, figure 2 indicates that Pt is added as particles and chloroplatinic acid. This inconsistency should be resolved.
Answer: Figure 2 has been modified in the revised manuscript according to the review’s suggestion.
- Figure 3. Clarify the features identified by a red arrow and ellipse in figure 3b. Derive the size of the Pt and TiO2 particles using the Scherrer equation and figure 3a data for comparison with other measurements.
Answer: The red arrow and ellipse in figure 3b indicate that Pt nanoparticles are deposited on TiO2. The size of Pt nanoparticles deposited on TiO2 particles calculated by Scherrer equation is about 2.05 nm.
Add labels to figure 3b ellipse (TiO2 particle?) and arrow (Pt nanoparticle?). Add a brief explanation to the text considering that the Pt particle size obtained by XRD (2.05 nm) is different than the value derived from TEM (~10 nm).
Answer: Figure 3b has been modified according to the review’s suggestion. The discrepancy for particle size may be caused by the measure condition and sample treatment of XRD and TEM.
- Section 3.3.1. It is stated that “It can be observed that evident aggregation and phase separation did not appear and Pt-TiO2 nanoparticles were uniformly distributed in the nanocomposite PEMs.”. What is expected if aggregation and phase separation occur? This information should be added to the text. It is also stated that “The rough morphology and partial agglomeration of the inorganic additives is most likely caused by the excessive Pt nanoparticles affecting the dispersion of TiO2 and FPGO in SPSU.”. This sentence is unclear considering that the amount of Pt loaded on TiO2 was not varied.
Answer: If aggregation and phase separation occur, obvious fibrous and granular crystals appear on the section. To avoid partial agglomeration of the inorganic additives, proper Pt was used to form Pt-TiO2 However, when the Pt-TiO2 content is increased up to 3 wt.%, some fibrous and granular crystals appeared on the cross section.
In section 2.4, second paragraph, indicate the weight ratio percentage of Pt-TiO2 nanoparticles to FPGO/SPSU corresponding to the 2.0 mg of Pt-TiO2
Answer: In section 2.4, “The membrane was dried at 40°C for 12 h to remove the residual solvent, followed by successively drying at 90°C for 12 h and vacuum drying at 60°C for further use.” has been modified as “The membrane was dried at 40°C for 12 h to remove the residual solvent, followed by successively drying at 90°C for 12 h. After vacuum drying at 60°C, the nanocomposite PEM with 0.1 wt.% Pt-TiO2 content was obtained.”
- Section 3.4.2, 1st It is stated that “The Ea values of all membranes which are between 15.31~22.96 kJ/mol by the Arrhenius plot are in the range of 14.3~39.8 kJ/mol by explaining the proton penetration behavior by Grotthuss mechanism.”. Clarify the meaning of this sentence. For instance, are the authors suggesting that for a purely Grotthuss mechanism the expected activation energy is in the range of 14.3 to 39.8 kJ/mol, which matches the experimentally determined range of 15.31 to 22.96 kJ/mol? Also, it is stated that “The significantly improved proton conductivity with elevated temperature can be attributed to the combined effect of the following two factors: (1) the increase of temperature will promote the movement of macromolecular chains and the absorption of water molecules in membrane materials, and (2) the increase of proton concentration due to the decrease of water dilution effect.”. However, the 2 listed explanations are seemingly contradictory. How can the water dilution effect be less important if more water is absorbed in the polymer? This contradiction needs to be resolved.
Answer: Section 3.4.2, the sentence “The Ea values of all membranes which are between 15.31~22.96 kJ/mol by the Arrhenius plot are in the range of 14.3~39.8 kJ/mol by explaining the proton penetration behavior by Grotthuss mechanism.” has been modified as “For a purely Grotthuss mechanism the expected activation energy is in the range of 14.3 to 39.8 kJ/mol, which matches the experimentally determined range of 15.31 to 22.96 kJ/mol.” In addition, the sentence “the increase of proton concentration due to the decrease of water dilution effect.” has been deleted in the revised manuscript to avoid some misleading.
The suggested change related to the Grotthuss mechanism was not implemented in the revised manuscript.
Answer: The suggested change related to the Grotthuss mechanism has been implemented in the revised manuscript.
- Figure 9. What is the original relative humidity or water content of the membrane? Conditions were clearly changed to create membrane dehydration. However, the method used is not described.
Answer: The whole test mould was placed in a thermostat at 60°C. The longer the test mould was in the thermostat, the less the water content in the membrane sample was. When the membrane sample reaches room temperature and 39% RH, the corresponding proton conductivity can be calculated according to formula (4) by measuring the impedance value of the membrane samples for every 5 min.
This explanation is still unclear. It might be much clearer to only say that the membrane holder was suddenly heated to 60°C while still in contact with ambient air (39 % RH) to induce dehydration, which was monitored by membrane conductivity measurements, assuming this is correct.
Answer: Section 3.4.2, 4th “The whole test mould was placed in a thermostat at 60°C. The longer the test mould was in the thermostat, the less the water content in the membrane sample was. When the membrane sample reaches room temperature and 39% RH, the corresponding proton conductivity can be calculated according to formula (4) by measuring the impedance value of the membrane samples for every 5 min.” has been modified as “The membrane holder was suddenly heated to 60°C while still in contact with ambient air (39% RH) to induce dehydration, which was monitored by membrane conductivity measurements.”
- It is stated that “Based on introducing GO polymer brushes as an inorganic additive and incorporating Pt-TiO2 nanoparticles fixed on polymeric PEM by forming cross-linked network structure, novel cross-linked PEM and nanocomposite PEMs with high performance and self-humidifying were prepared to solve leakage of inorganic additives during use and compatibility problem with organic polymers.”. The authors cannot comment on the leakage of inorganic additives because this aspect was not quantitatively evaluated. Although GO was cross-linked, the Pt-TiO2 particles are only maintained by weak hydrogen bonds.
Answer: There may be no leakage of inorganic additives because Pt-TiO2 nanoparticles were not observed according to the cross-section SEM images of the membrane samples in the paper.
This statement is still misleading because the leakage of inorganic additives was not measured using an actual fuel cell. It would be preferable to mention this clearly and add that there is an indication that the leakage of inorganic additives may not occur based on SEM images.
Answer: The sentence “Although the leakage of inorganic additives was not measured using an actual fuel cell, there is an indication that the leakage of inorganic additives may not occur based on SEM images.” has been added into the revised manuscript of the conclusion section.